# Effectiveness of Sex Education in Adolescents

**Aroa Cortínez-López** [1]**, Daniel Cuesta-Lozano** [2,*]  **and Raquel Luengo-González** [2]

1    Obstetric Unit, Clinic University of Navarra, 28027 Madrid, Spain; aroa.cortinez@edu.uah.es
2    Nursing and Physiotherapy Department, University of Alcala, Alcalá de Henares, 28805 Madrid, Spain; raquel.luengo@uah.es
*    Correspondence: daniel.cuesta@uah.es

**Abstract:** Adolescence is the time during which the personal and sexual identity develops. The specific characteristics of adolescents and the lack of maturity facilitates the acquisition of sexual risk behaviors such as relaxation in the use of barrier contraceptives or the use of toxic substances, alcohol or drugs during sexual relations, increasing of sexually transmitted infections (STIs) and unwanted pregnancies. Health education in sexuality is one of the best ways to prevent risk behaviors and to promote healthy and responsible sexuality. While there have been different works in sex education in adolescents, there is still a lack of a comprehensive systematic literature review that including randomized and non-randomized clinical trials and quasi-experimental pre-post studies addressing education programs that provide information on healthy and responsible sexuality. Further, it is noted that a protocol drafted in consideration of the existing approaches is needed to present a basis for a systematic literature review in this area. This article, therefore, proposes a review protocol that will evaluate the impact of comprehensive sex education programs in the level of knowledge about STIs, behavioral level concerning the frequency of use of effective contraceptive methods and level of knowledge about sexual identity, diversity and/or responsible sexuality, after the intervention.

**Keywords:** adolescent; risk behaviors; sexual; health education





## 1. Introduction

### 1.1. Description of the Situation

Sexuality is a central aspect throughout humans' life. The adolescence is the time during which the personal and sexual identity develops [1]. Sexuality takes into account aspects such as sex, gender identity, gender roles, sexual orientation, eroticism and intimacy. It is influenced by various factors like biology (age and sex), society (culture and religion) and psychology [2].

The majority of the human population has their first sexual encounters during their adolescence. The specific characteristics of adolescents (need for independence, impulsiveness and emotional lability) and the lack of maturity at the time of initiating sexual relations facilitates the acquisition of risk behaviors in these relationships, among which the following stand out as the most prevalent [3,4]:

- Sexual relations are usually sporadic and in unsuitable spaces such as cars and parks;
- Quickly changing sexual partners;
- The idea of invulnerability with a false perception that the probability of contracting a sexually transmitted infection (STI) is very small;
- Relaxation in the use of barrier contraceptives, to avoid rejection by not accepting a sexual relationship if there is no condom or if the sexual partner does not want to use it;
- The use of toxic substances, alcohol or drugs, decreases the attention in the use of protective measures, control and voluntariness of the relationships.

In Spain in 2019 the fertility rate in adolescents between 15 and 19 years old was 6.07 births per thousand women. In total, there were 7094 births to teenage mothers,

making up for 1.9% of the births that year [5]. In Spain in 2017 the rate of voluntary interruptions of pregnancy was 8.84 per thousand women under 19 years of age and 17.42 per thousand women between the ages of 20 and 24 [6].

STIs constitute a group of pathologies of diverse etiology, in which the sexual route is the main form of contagion, including vaginal, anal and oral sex. STI agents can also be spread by non-sexual means, e.g., blood transfusions, tissue transplants or from mother to child during pregnancy and childbirth [7].

The World Health Organization (WHO) estimates that more than 1 million people contract an STI every day [8]. In 2008, 498.9 million new cases were counted in adults aged 15–49, which is an 11% increase from 2005. Only considering gonorrhea, syphilis, chlamydia and trichomonas, which constitute the four most frequent curable STIs, 2016 saw 376 million new cases worldwide in patients between 14 and 49 years of age, of which, 46.8 million cases happened in the European Union. The pathologies that have increased their prevalence the most are gonorrhea and trichomonas [9,10].

STIs are a major public health concern. If they are not diagnosed and treated early, the resulting complications and sequels can be very serious. A large proportion of the infections are asymptomatic. This means that they are under-diagnosed and under-treated, which in many cases entails the contagion of sexual partners and later diagnosed pathologies such as: infertility, cervical cancer and pelvic inflammatory disease (PID) [7,8]. The most frequently appearing symptoms leading to the diagnosis of an STI are abnormal vaginal discharge, genital ulcers or non-ulcerative lesions and pelvic pain [7,8].

### 1.2. Description of the Intervention

The WHO defines sexual health as: "a state of physical, emotional, mental and social well-being concerning sexuality; it is not merely the absence of disease, dysfunction or weakness. Sexual health requires a positive and respectful approach to sexuality and sexual relationships, as well as the possibility of having pleasurable and safe sexual experiences, free of coercion, discrimination and violence" [11].

Health education is a primary prevention method that is effective in promoting behavioral changes and the acquisition of healthy habits. Health education aims to increase the population's knowledge of their health and to provide different methods to promote health through various activities [11].

To achieve adequate sexual health, sex education programs are necessary to transmit knowledge, skills and values to adolescents so that they can establish respectful and healthy social and sexual relationships, become aware of how their decisions can affect them and the people around them and become aware of their rights and watch over them. This can be achieved through comprehensive sex education (CSE), which promotes healthy sexuality by addressing a variety of topics such as: puberty, sexual identity, sexual orientation, healthy relationships, STIs and contraception [11,12].

The majority of adolescents demand sexual education. Unfortunately, this request is often not taken into account or the provided CSE is distorted, partial or executed with an excessive emphasis on biological aspects of reproduction and lacking the focus on the acquisition of healthy behaviors [11].

For fruitful CSE to happen the appropriate communication channel must be chosen. These can be: talks in institutes, group discussions or individualized talks. Furthermore, the information has to be appropriate for the cultural and age characteristics of the participants. Health education should be based on multidisciplinary work with the family, teachers, sexologists and health professionals [13].

### 1.3. How the Intervention Could Work

Proper CSE can be the cornerstone in the effort towards the acquisition of healthy sex habits and the prevention of STIs. It has been shown that CSE can delay the onset of sexual relations, decrease the number of sexual partners, reduce risky sexual activity and increase the use of contraceptives [11].

Various studies on the impact of health education on STIs show that there is an increase in the level of knowledge about STIs and contributes to the reduction of false beliefs regarding STIs [14–16]. The effectiveness of these interventions is greater if sexual relations have not yet begun [15,16]. However, no statistically significant differences have been described concerning rates of unwanted pregnancy, STI incidence and sexual health promotion [17].

Sex education must not only focus on STIs and their prevention. Different studies show that by providing a more comprehensive vision of sexuality, adolescents will make better decisions regarding sexuality, equality and respect for diversity will be favored, gender violence will be prevented and it will allow them to enjoy healthier and more responsible sexuality [11,12].

In general, health education programs for adolescents on sexuality increase the tendency to research further questions in this regard using adequate sources such as: families, teachers and health care professionals [16].

Finally, studies show that such programs do not lead to an increase in sexual practice, they naturally increase with age, regardless of whether the adolescents have been present at an intervention or not [14].

### 1.4. Why It Is Important to Do This Review

Health education in sexuality is one of the best ways to prevent STIs and to promote healthy and responsible sexuality. This review will examine interventions made in the educational context based on CSE.

There is scientific evidence examining educational strategies to improve the use of contraceptive methods and prevent STIs [18,19]. However, there are no studies on interventions that meet the WHO-defined criteria for correct comprehensive sexuality education [11]. Therefore, in addition to evaluating educational programs on STIs and contraception, will be analyzed educational programs that address topics such as puberty, sexual identity, sexual diversity and healthy relationships.

### 1.5. Objectives

To evaluate the effect of CSE interventions on adolescents.

## 2. Materials and Methods

### 2.1. Study Selection Criteria

#### 2.1.1. Type of Studies

This review will be included randomized clinical trials by individuals and by groups, non-randomized clinical trials and quasi-experimental pre-post studies.

#### 2.1.2. Type of Participants

This review will use studies carried out with adolescents (19 years old or younger). Generally, the interventions carried out in educational centers are organized by academic courses and not by specific age groups. In addition, high schools may have students over 19 years of age, so in this review, most participants will be under 19. The lower age limit will be set at 12 years, since, although different organizations consider that adolescence begins at 10 years of age [20], in secondary education adolescents under 12 years are not commonly found. Studies with participants of any nationality, gender and sexual orientation will be included.

#### 2.1.3. Type of Intervention

Interventions carried out mainly in secondary schools intending to promote sexual health will be studied. However, interventions carried out in the general community will also be considered, but most of the interventions must have been carried out in high school.

Interventions carried out at the university, primary schools or special education schools will not be included, because the structure and content of the interventions may

differ greatly as there are significant age differences to the type of population selected for the review. Those interventions not carried out by health professionals or teaching professionals will be excluded.

Studies with education programs that provide information on healthy and responsible sexuality will be included in this review. They should also address different STIs and the various methods of preventing them. Furthermore, they should provide information on one or more contraceptive methods, focusing on those considered most effective in preventing STIs and unwanted pregnancies. They may also have addressed issues of sexual identity and changes during adolescence.

2.1.4. Type of Result Measurement

Main results:

Studies evaluating one or more of the following outcomes will be included:

- The level of knowledge about STIs and STI prevention;
- The level of knowledge about changes in adolescence, sexual identity, sexual diversity and/or responsible sexuality;
- The behavioral level with regard to the frequency of use of effective contraceptive methods to prevent STIs and unwanted pregnancy, after the intervention.

The level of knowledge can be assessed through validated scales administered pre-intervention and at least 3 months after the intervention and for more reliable results, 6 months after the intervention.

For behavioral changes, they should assess use and adherence to contraceptives effective in preventing STIs in the last 3 months (minimum) or use of contraception during the last intercourse.

Secondary outcomes:

These measures can complement the primary outcomes specified above:

- Health-seeking behaviors, such as performing Pap smears, increasing STI testing and treatment;
- Behaviors that target healthy sexuality, assessing the number of sexual partners, use of alcohol during sexual practice and the perception of the risk of contracting an STI.

The time frame for evaluation of these items should be 3 months and for high-quality evidence, it should be 6 months.

*2.2. Search Methods for Identifying Studies*

A search of bibliographic databases will be conducted. Non-electronic journals will not be searched. Articles must have been published within the last 5 years and must be written in English or Spanish.

The following databases will be used for the search:

1. PubMed;
2. CENTRAL (Cochrane Central Register of Controlled Trials);
3. ERIC (Educational Resources Information Center);
4. EMBASE (Excerpta MedicaDatabase);
5. CINAHL (Cumulative Index to Nursing & AlliedHealth Literature);
6. Cuiden.

Appendix A includes the search strategies used.

*2.3. Data Extraction and Analysis*

2.3.1. Selection of Studies

All titles and abstracts identified by the literature search will be included. Two authors will review the studies obtained to assess whether or not they are relevant to the review. Studies that seem appropriate will be reviewed by two authors by examining the full text. In case of disagreement, they will be resolved by discussion and, if considered necessary, a third author will analyze the studies to assess whether or not they are appropriate.

### 2.3.2. Data Extraction and Management

Data extraction will be performed by two authors. One author will enter the obtained data into Review Manager (RevMan 2014) and a second author will review whether or not this data has been entered correctly. The data will include:

- Basic characteristics of the study: authors, year and journal;
- Characteristics of the participants: sample size, sex, mean age, nationality and loss rate;
- Characteristics of the intervention: the place of performance, content, activities, frequency and duration of the intervention;
- Evaluation of the intervention: scales used, time of the evaluation and results obtained after it.

### 2.3.3. Assessment of the Risk of Bias in the Included Studies

The risk of bias of each study will be assessed using the different tools described in the Cochrane Manual for Systematic Reviews of Interventions. For each study, two reviewers will independently assess the probability of bias in the following domains [21]:

- Selection (sequence generation and allocation concealment);
- Implementation (blinding of participants and staff);
- Detection (blinding of outcome evaluators);
- Wear and tear (incomplete outcome data);
- Notification (selective notification of results);
- Dropout (general and group dropout rates);
- Other sources of bias (recruitment bias or baseline imbalance bias).

For each domain, studies will be classified as "high", "low" or "unclear risk" bias. In case of disagreement, it will be resolved by discussion and if necessary by a third author.

### 2.3.4. Evaluation of the Quality of the Studies

The Quality Assessment Tool for Quantitative Studies (Effective Public Health Practice Project 2007) will be used to assess the quality of the studies.

### 2.3.5. Measures of Intervention Effect

The Odds Ratio (OR), using the Mantel-Haenszel method, with an 85% confidence interval (CI), will be used for the analysis of the results of each study with dichotomous data. For continuous results, the mean difference (MD) with standard error will be used. If studies used different scales to measure the same outcome, the standardized mean difference (SMD) will be applied.

For studies that analyze changes from a baseline, the mean differences will be included as a covariate in a regression model or covariance analysis (ANCOVA). The combined standard deviation (SD) of the intervention and post-intervention control SD will be calculated.

### 2.3.6. Problems Related to the Unit of Analysis

In the case of group randomized trials (institute classrooms) if the analysis is done at the group level no special statistical analysis is required. However, if the studies report on an individual level, it will be assessed whether the author has taken into consideration the effect of clustering and using the indicated statistical techniques as a "multilevel model".

### 2.3.7. Lack of Data

If any statistical data relevant to the review is missing or unclear, the author will be contacted to request such data.

### 2.3.8. Evaluation of Heterogeneity

A meta-analysis will not be performed due to the wide variability of educational interventions and the different measures that can be used to assess the outcome of such interventions. Statistical tests of heterogeneity will not be carried out, but heterogeneity will

be taken into account in the design of the studies, the population studied, the interventions carried out and the way of measuring the results.

2.3.9. Identification of Publication Bias

The publication bias will be evaluated through the funnel plot, in which the sample size of each job is represented against the size of the detected effect. This method usually presents effect sizes plotted against their standard errors or precisions (the inverse of standard errors). It is recommended to complement the funnel plot with statistical techniques such as the Begg or Egger test [21,22]. The Begg rank test examines the correlation between the effect sizes and their corresponding sampling variances; a strong correlation implies publication bias. Additionally, Egger's test regresses the standardized effect sizes on their precisions; in the absence of publication bias, the regression intercept is expected to be zero [22].

## 3. Conclusions

During adolescence, a large part of sexual relations is initiated. Due to the characteristics of this stage, the acquisition of risky sexual behaviors is common. For this reason, comprehensive sex education programs should be started during adolescence.

Sex education on STIs and contraceptives is effective in increasing knowledge, but this increase is not always related to changes in sexual practices. This review will be assessed studies with education programs that not only focus on STIs and contraceptives but also address topics such as healthy sexuality, self-esteem, changes in adolescence and sexual diversity. These programs should be delivered by health professionals or teachers. The objective of this review protocol is to find how these programs affect the knowledge of sexuality and if they improve practices related to sexuality, promoting attitude change and the pursuit of healthy sexuality.

**Author Contributions:** Conceptualization, A.C.-L.; methodology, A.C.-L.; writing-original draft preparation, A.C.-L.; review and editing, R.L.-G. and D.C.-L.; visualization, R.L.-G.; supervision, D.C.-L. and R.L.-G.; project administration, D.C.-L. All authors have read and agreed to the published version of the manuscript.

**Funding:** This research received no external funding.

**Institutional Review Board Statement:** Not applicable.

**Informed Consent Statement:** Not applicable.

**Data Availability Statement:** Since this is a systematic review protocol, Data Availability Statement is not yet available.

**Conflicts of Interest:** The authors declare no conflict of interest.

## Appendix A

*Search Methods*
Pubmed and Cuiden:
("Health Education" OR "Sex Education") AND ("Contraceptive Agents" OR "Sexually Transmitted Diseases" OR "Sexual Behavior") AND ("Adolescent" OR "Young Adult") AND (School OR Students) NO ("Education, Non-Professional")
Dates of publication: 5 years
Languages: English or Spanish
Cochrane Library:
Title, abstract, keywords: ("Health education" OR "Sex education")
And Title, Summary, Keywords: ("Contraceptive Agents" OR "Sexually Transmitted Diseases" OR "Sexual Behavior")
And Title, summary, keywords: ("teenager" OR "young adult")
And Title, summary, keywords: (school OR students)
NO Title, summary, keywords: (Education, non-professional)

ERIC:

("Contraceptive Agents" OR "Sexually Transmitted Diseases" OR "Sexual Behavior") AND ("Adolescent" OR "Young Adult") AND (School OR Students) NO ("Education, Non-Vocational")

Type of publication: Reports—Research

Descriptor: Sex education

Keyword: Teenagers

Web of Science (other databases: EMBASE and CINAHL):

SUBJECT: ("Health Education" OR "Sex Education") AND SUBJECT: ("Birth Control Agents" OR "Sexually Transmitted Diseases" OR "Sexual Behavior") AND SUBJECT: ("Adolescent" OR "Young Adult") AND SUBJECT: (School OR Students) NOT SUBJECT: ("Education, Non-Vocational")

Refined by: Duration: 5 years

Search language: English or Spanish

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
