# Peer review of "Effectiveness of Sex Education in Adolescents"

_sexes, doi:10.3390/sexes2010012_

Round 1

Reviewer 1 Report

The paper is not prepared for blind review.

Check commas (for instead before "and"/"or": it's a systematic fault in the paper) and hyphenation at end of line.
Expressions like "life of humans" sound strange. Better "human life".

Avoid 1st person: write in an unpersonnal style.

"Multiple sexual partners are very frequent (2 or more)" is an ambiguous sentence. All at the same time? Whatever, sure that it is "very" frequent?

Results and discussion are missing.

Author Response

Thank you very much for your comments as they will allow me to improve the quality of the article.

Point 1: The paper is not prepared for blind review.

Response 1: Indeed, the paper is not prepared for a blind review, we have followed the template that was sent to us to enter the article and in it, we were instructed to put the data of the author.

Point 2: Check commas (for instead before "and"/"or": it's a systematic fault in the paper) and hyphenation at end of line.
Expressions like "life of humans" sound strange. Better "human life".

Response 2: We have corrected the commas and hyphenation at end of line. We have also changed the expression you mentioned.

Point 3: Avoid 1st person: write in an unpersonnal style.

Response 3: We have removed from the text the writing in first person.

Point 4: "Multiple sexual partners are very frequent (2 or more)" is an ambiguous sentence. All at the same time? Whatever, sure that it is "very" frequent?

Response 4: We have modified the phrase "Multiple sexual partners are very frequent (2 or more)" so that it does not sound so ambiguous.

Point 5: Results and discussion are missing.

Response 5: Because this article is from a systematic review protocol the results are not yet available, but it will be a pleasure to share them with you once the review is completed.

Again, thank you very much for your recommendations.

Reviewer 2 Report

The paper's objective is interesting. However, the paper is still in planning stage and there are multiple areas of improvements. First, Line 110: "... that meet the criteria of adequate comprehensive sexuality education....". Now, "adequate" is a subjective term. Thus, the authors may consider clarifying how this is defined. Regarding approach of the systematic review, it is good that the paper clarifies the step wise approach. Also, the selection mentions of both English and Spanish articles, which is good. However, may clarify why such choices are made. Are the points suggested in Lines 215 to 221 based on any other paper that guides avoiding biases in systematic review with multiple researchers involved? If so, please cite accordingly. Identification of publication bias can be explained further, especilaly for readers across disciplines. Overall, the suggested framework for a future study sounds interesting. However, it will possibly be more interesting to see the review outcomes than the plan. There are already some papers in this area; and such can be considered to give an indication as to the level of depth the review is expecting to cover once the actual data are obtained.

Author Response

Thank you very much for your comments as they will allow me to improve the quality of the article.

Point 1: First, Line 110: "... that meet the criteria of adequate comprehensive sexuality education....". Now, "adequate" is a subjective term. Thus, the authors may consider clarifying how this is defined.

Response 1: We have corrected the term "adequate" so that it was not a subjective expression.

Point 2: The selection mentions of both English and Spanish articles, which is good. However, may clarify why such choices are made. 

Response 2:  On the other hand, we have selected both languages since all the authors speak both Spanish and English.

Point 3: Are the points suggested in Lines 215 to 221 based on any other paper that guides avoiding biases in systematic review with multiple researchers involved? If so, please cite accordingly. Identification of publication bias can be explained further, especilaly for readers across disciplines.

Response 3: We have also added the bibliographic reference in the section on the biases of the review and we have expanded the explanation of the publication bias, as you have indicated.

Point 4: Overall, the suggested framework for a future study sounds interesting. However, it will possibly be more interesting to see the review outcomes than the plan.

Response 4: Because this article is from a systematic review protocol the results are not yet available, but it will be a pleasure to share them with you once the review is completed.

Again, thank you very much for your recommendations.

Reviewer 3 Report

This is a very interesting and promising study. I cannot wait to read the final review results. I have three suggestions.

1. The authors mentioned that they will "assess studies with education programs that not only focus on STIs and contraceptives but also address topics such as healthy sexuality, self-esteem, changes in adolescence, and sexual diversity." However, the literature review primarily focuses on how sexual education can help reduce STIs and promote contraceptive use. I'd suggest the authors adding more information about why other sexual education goals (e.g., healthy sexuality, self-esteem, and sexual diversity) are important to address and how sexual education can help achieve those goals.

2. Some paragraphs are very short and could have been combined into longer paragraphs (e.g., line 43-48, line 65-67). 

3. Line 141-151 would be more appropriate in the literature review. Also, I'd suggest the authors using more paraphrase than direct quotes.

Author Response

Thank you very much for your comments as they will allow me to improve the quality of the article. It will be a pleasure to share with you the results once we finalize the review.

Point 1: The authors mentioned that they will "assess studies with education programs that not only focus on STIs and contraceptives but also address topics such as healthy sexuality, self-esteem, changes in adolescence, and sexual diversity." However, the literature review primarily focuses on how sexual education can help reduce STIs and promote contraceptive use. I'd suggest the authors adding more information about why other sexual education goals (e.g., healthy sexuality, self-esteem, and sexual diversity) are important to address and how sexual education can help achieve those goals.

Response 1: We have added more information on the importance of addressing more topics in sex education as you have indicated.

Point 2: Some paragraphs are very short and could have been combined into longer paragraphs (e.g., line 43-48, line 65-67). 

Response 2: We have also combined very short paragraphs. 

Point 3: Line 141-151 would be more appropriate in the literature review. Also, I'd suggest the authors using more paraphrase than direct quotes

Response 3: And we have removed from the methodology and added in the literature review the paragraphs you have indicated to us, also changing a part of the direct quotes to paraphrases.  

Again, thank you very much for your recommendations.

Round 2

Reviewer 1 Report

"Multiple sexual partners are very frequent (2 or more)" is an ambiguous sentence: the author says that it is reviewed, but it isn't.

Author Response

Thank you very much for your contribution,

Point 1: "Multiple sexual partners are very frequent (2 or more)" is an ambiguous sentence: the author says that it is reviewed, but it isn't.

Response 1: Sorry for the inconvenience, the sentence change we made to replace it with a less ambiguous one must not have been saved well. 
In the new manuscript, that phrase is eliminated and in its place, we have put another less ambiguous one.

Thank you very much.

Reviewer 2 Report

Since there is no result yet, it is still just a plan. In case such a paper is acceptable for the publication, yes this certainly a better and clearer version, and appreciate the authors' work concerning this. I would suggest some improvements concerning abstract to clarify to the reader that this will be a study will be undertaken for the stated purpose 1.4.

In abstract, rather than saying "A systematic review was proposed …", consider rephrasing like:

"While there have been different works in sex education in adolescents, there is still a lack of a comprehensive systematic literature review that …… Further, it is noted that a protocol drafted in consideartion of the existing approaches is needed to present a basis for a systemtic literature review in this area. This article, therefore, proposes a review protocol ...".

Or, something similar. The main point is the basis of this paper needs to be more explcit if it is to be published as a plan paper. A reader reads a published article not only to know what the respective researchers have doen or will do, but also to enhance his/her knoweldge of the domain and possibly conducting a research as a follow up at his/her end also. Thus, few sentences to that effect will be good - just a  thought for the authors to consider.

Other than that, on Line 258, the tests can be cited: "... Begg or Egger test (citataion)"

Lastly, on lines 272-274, "It is expected to find …": perhaps this shows some conjecture already formed before the review has been undertaken. Possibly it is better to avoid this, ands can pitch the same as a sort of objective of conducting this review, which I guess will come as a follow up.

Author Response

Thank you very much for your contribution,

Point 1: In abstract, rather than saying "A systematic review was proposed …", consider rephrasing like:...

Response 1: We have modified the summary according to your instructions.

Point 2: Other than that, on Line 258, the tests can be cited: "... Begg or Egger test (citataion)"

Response 2: The citation you requested has been added.

Point 3: Lastly, on lines 272-274, "It is expected to find …": perhaps this shows some conjecture already formed before the review has been undertaken. Possibly it is better to avoid this, ands can pitch the same as a sort of objective of conducting this review, which I guess will come as a follow up.

Response 3: The conclusion has been corrected following your recommendations.

Thank you very much.